# Discovery of Two Inhibitors of the Type IV Pilus Assembly ATPase PilB as Potential Antivirulence Compounds

Keane J. Dye,[a] Nancy J. Vogelaar,[b,c] Megan O'Hara,[a] Pablo Sobrado,[b,c,d] Webster Santos,[b,d,e] Paul R. Carlier,[b,e] Zhaomin Yang[a,b,d]

[a]Department of Biological Sciences, Virginia Tech, Blacksburg, Virginia, USA

[b]Virginia Tech Center for Drug Discovery, Virginia Tech, Blacksburg, Virginia, USA

[c]Department of Biochemistry, Virginia Tech, Blacksburg, Virginia, USA

[d]Virginia Tech Center for Emerging, Zoonotic and Arthropod-borne Pathogens, Virginia Tech, Blacksburg, Virginia, USA

[e]Department of Chemistry, Virginia Tech, Blacksburg, Virginia, USA

**ABSTRACT** With the pressing antibiotic resistance pandemic, antivirulence has been increasingly explored as an alternative strategy against bacterial infections. The bacterial type IV pilus (T4P) is a well-documented virulence factor and an attractive target for small molecules for antivirulence purposes. The PilB ATPase is essential for T4P biogenesis because it catalyzes the assembly of monomeric pilins into the polymeric pilus filament. Here, we describe the identification of two PilB inhibitors by a high-throughput screen (HTS) *in vitro* and their validation as effective inhibitors of T4P assembly *in vivo*. We used *Chloracidobacterium thermophilum* PilB as a model enzyme to optimize an ATPase assay for the HTS. From a library of 2,320 compounds, benserazide and levodopa, two approved drugs for Parkinson's disease, were identified and confirmed biochemically to be PilB inhibitors. We demonstrate that both compounds inhibited the T4P-dependent motility of the bacteria *Myxoccocus xanthus* and *Acinetobacter nosocomialis*. Additionally, benserazide and levodopa were shown to inhibit *A. nosocomialis* biofilm formation, a T4P-dependent process. Using *M. xanthus* as a model, we showed that both compounds inhibited T4P assembly in a dose-dependent manner. These results suggest that these two compounds are effective against the PilB protein *in vivo*. The potency of benserazide and levodopa as PilB inhibitors both *in vitro* and *in vivo* demonstrate potentials of the HTS and its two hits here for the development of anti-T4P chemotherapeutics.

**IMPORTANCE** Many bacterial pathogens use their type IV pilus (T4P) to facilitate and maintain an infection in a human host. Small-molecule inhibitors of the production or assembly of the T4P are promising for the treatment and prevention of infections by these bacteria, especially in our fight against antibiotic-resistant pathogens. Here, we report the development and implementation of a method to identify anti-T4P chemicals from compound libraries by high-throughput screen. This led to the identification and validation of two T4P inhibitors both in the test tubes and in bacteria. The discovery and validation pipeline reported here as well as the confirmation of two anti-T4P inhibitors provide new venues and leads for the development of chemotherapeutics against antibiotic-resistant infections.

**KEYWORDS** antivirulence, PilB ATPase, type IV pilus, T4P assembly, high-throughput screen, benserazide, levodopa

Address correspondence to Zhaomin Yang, zmyang@vt.edu.

The authors declare no conflict of interest.

The bacterial type IV pilus (T4P), an extracellular protein filament (1–5), is one of the most critical and prevalent structures or components essential for the pathogenesis of bacterial pathogens (6–9). Such components are known as virulence factors and function as weapons and armor to aid and abet bacterial pathogens to initiate, establish, and/or sustain the infection of a host. Their elimination or inhibition is known to abrogate or attenuate the ability of a pathogenic bacterium to cause infections. As an extracellular structure, the T4P is known to facilitate bacterial adherence to the surfaces

of host cells or medical implants to initiate colonization (10, 11). In some cases, development of a biofilm may ensue, resulting in a chronic and persistent infection that is recalcitrant to antibiotic chemotherapy. Bacteria which possess and use the T4P as a virulence factor include prominent pathogens that have become increasingly antibiotic resistant. The World Health Organization (WHO) has published a priority list of bacterial pathogens requiring new therapeutics because of their antibiotic resistance (12). This list includes *Acinetobacter baumannii*, *Pseudomonas aeruginosa*, *Neisseria gonorrhoeae*, and *Haemophilus influenzae*, all of which are known to produce T4P (8, 9, 13–15). This underscores the importance of developing anti-T4P therapies against bacterial infections, especially those with increasing resistance to antibiotics.

The formation and proper function of the T4P relies on a dozen conserved T4P or Pil proteins, including the PilB ATPase (16). Most of the Pil proteins form a supramolecular structure known as the T4P nanomachine or T4P machinery (T4PM). The dynamic T4PM functions to construct the T4P filament and anchor it to the bacterial cell body. The pilus filament is a polymer of pilins (PilA). Before being in the polymer, pilins exist as monomers imbedded in the cell membrane. PilB is the cytoplasmic ATPase that hydrolyzes ATP to provide energy for the assembly of the T4P filament from monomeric pilins (1, 16–18). In some cases, the assembled T4P can be disassembled or retracted by the ATPase PilT. The recurrent cycles of T4P assembly/extension and disassembly/retraction are known to power twitching motility in bacteria such as *P. aeruginosa*, *N. gonorrhoeae*, and *Acinetobacter* spp., as well as the social (S) motility in *Myxococcus xanthus* (19–22). This T4P-dependent surface motility can be examined by measuring colony expansion on agar surfaces, providing a convenient bioassay for the functions of T4P. The link between bacterial virulence and the T4P or piliation has been observed with *pilT* and *pilB* mutants. *pilT* mutants are hyperpiliated because they can assemble, but cannot retract, their T4P. This hyperpiliation has been associated with the elevated transcription of virulence genes and enhanced biofilm production (8, 23, 24). In contrast, *pilB* mutants are not piliated, and they are either avirulent or attenuated, suggesting that chemical inhibitors of PilB hold promise for the development of antivirulence therapies.

There have been three reports on the discovery of small molecules with anti-T4P activity. The compounds identified in these three studies were trifluoperazine (25), P4MP4 (1-[(piperidin-4-yl)methyl]piperidin-4-ol) (26, 27) and quercetin (28). The anti-T4P activity of the antipsychotic drug trifluoperazine and related phenothiazines was accidentally discovered because of their ability to disaggregate *Neisseria meningitidis* cell clusters (25). Using cells and a humanized mouse model, phenothiazines were found to reduce bacterial colonization, bacteria-induced cell injury, and vascular lesions. In a mouse infection model, these compounds were found to provide adjunctive benefits with antibiotics. Genetic studies identified the $Na^+$ pumping NADH-ubiquinone oxidoreductase ($Na^+$-NQR) complex as the target of phenothiazines. P4MP4 was identified from a library of 2,239 compounds by a high-throughput screen (HTS) based on the reduction of *N. meningitidis* adhesion to cultured cells (26). It was subsequently found that P4MP4, reminiscent of phenothiazines, affected *N. meningitidis* cellular aggregation and T4P levels. Biochemical studies suggested that the target of P4MP4 is the *N. meningitidis* T4P assembly ATPase. In comparison, quercetin was identified as a PilB inhibitor *in vitro* by a HTS because it diminished the binding of *Chloracidobacterium thermophilum* PilB (*Ct*PilB) to a fluorescent ATP analogue (28). Further examination indicated that it reduced T4P-dependent motility and T4P assembly in the model bacterium *M. xanthus*. These three studies show that T4P assembly and PilB are valid targets for further exploration of the antivirulence strategy.

Here, we report the identification of two PilB inhibitors in *vitro* and the validation of their anti-T4P activities *in vivo*. The inhibitors are benserazide and levodopa, two structurally related compounds approved for combinatorial therapy of Parkinson's disease (PD). We initially identified these inhibitors from a library of 2,320 compounds in a HTS against *Ct*PilB using an ATPase-based assay. We demonstrate that both compounds inhibit PilB *in vitro* by interacting with its highly conserved C-terminal ATPase domain. The results indicate that benserazide and levodopa are effective *in vivo* because they inhibit the T4P-dependent motility of *M. xanthus* and *Acinetobacter nosocomialis*, an opportunistic pathogen. We further show that they inhibit T4P assembly in *M. xanthus* and T4P-dependent biofilm formation

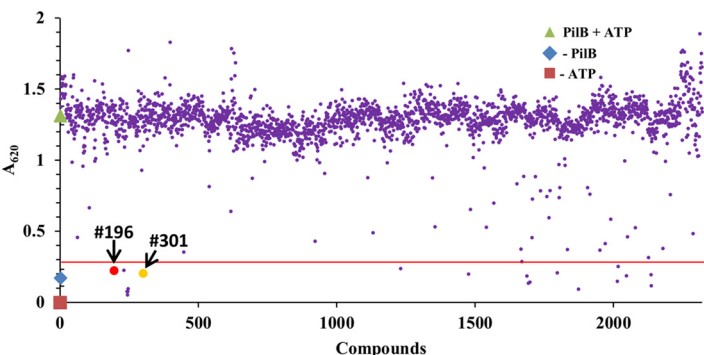

**FIG 1** A high-throughput screen (HTS) identified benserazide and levodopa as PilB inhibitors. The compound library was screened as described in the text. On the *y* axis, green triangle represents the reading from the control with PilB, ATP, and solvent (PilB+ATP); blue diamond represents that without PilB (−PilB); and red square represents that without ATP (−ATP). Red line indicates 90% inhibition of PilB activity. Compounds 196 and 301 are benserazide and levodopa, respectively.

in *A. nosocomialis*. These results highlight the potential of both benserazide and levodopa as leads for the development of anti-T4P chemotherapeutics and validate the approach of using *Ct*PilB as a model protein to identify PilB inhibitors for antivirulence purposes.

## RESULTS

**Benserazide and levodopa were identified as PilB inhibitors in HTS.** We first examined whether the malachite green (MG)-based ATPase assay for *Ct*PilB (29, 30) could be adapted for a HTS. In this assay, phosphates from ATP hydrolysis and the MG reagents form green complexes that can be quantified calorimetrically with a plate reader. Using 384-well microplates, the PilB ATPase assay was optimized with respect to enzyme concentration, length of incubation, and ATP concentration. The optimized and finalized conditions are described in Materials and Methods. To validate the HTS assay, the Selleckchem compound library of 273 kinase inhibitors was screened in a pilot experiment which identified quercetin, a known PilB inhibitor from our previous study (28).

Using the HTS protocol, the MicroSource SPECTRUM library of 2,320 compounds was screened with the results shown in Fig. 1. Three sets of controls were included in multiple wells in the screen: the first set contained PilB and ATP (PilB+ATP) with dimethyl sulfoxide (DMSO, the solvent for the library compounds), while the second and third sets replaced PilB (−PilB) or ATP (−ATP) with buffers. The screen was conducted with eight 384-well plates. Based on the controls with and without PilB, the average Z′ factor (31) from the eight screen plates was calculated to be 0.57, confirming a good and reliable assay for the screen. There were 20 compounds that inhibited *Ct*PilB ATPase activity by 90% in this screen (Fig. 1). The manuscript focuses on benserazide and levodopa, library compounds 196 and 301, respectively. These are two structurally related drugs (Fig. 2) approved for combination therapies of PD (32). Because these two compounds are approved for clinical use, there is a wealth of medically relevant information on them (33, 34).

**Benserazide and levodopa are PilB inhibitors interacting with the ATPase domain.** To confirm the inhibitory effects of benserazide and levodopa, we examined the dose response of the PilB ATPase to both compounds using the MG-based assay in benchtop experiments (Fig. 2). Our results indicated that the activity of PilB as an ATPase decreased with increasing concentrations of benserazide and levodopa. Benserazide appeared to be a more potent inhibitor than levodopa. The former reduced PilB activity to a basal level at concentrations of $\geq 32$ $\mu$M with a calculated 50% inhibitory concentration (IC$_{50}$) of 2.1 $\pm$ 0.4 $\mu$M (Fig. 2A). The latter required a concentration of $>512$ $\mu$M to do the same, with a calculated IC$_{50}$ of 58 $\pm$ 13 $\mu$M (Fig. 2B). Notwithstanding the differences in potency, these results confirmed the validity of the activity-based HTS because both benserazide and levodopa are genuine inhibitors of the *Ct*PilB ATPase.

To probe the mechanisms of inhibition, we examined benserazide and levodopa for their effects on the binding of PilB to the fluorescent ATP analog MANT-ATP (28). As shown

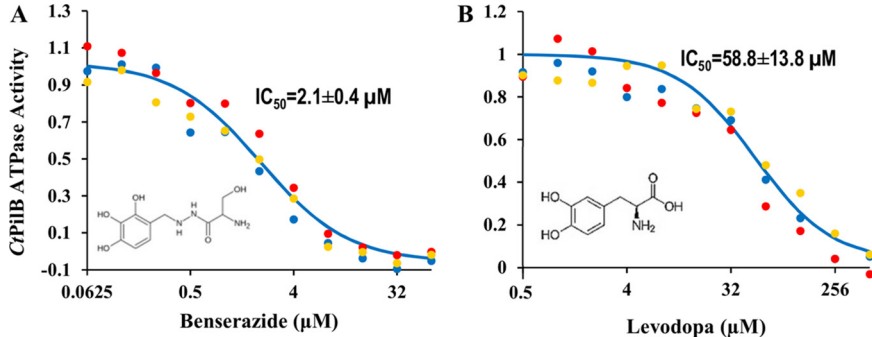

**FIG 2** Benserazide and levodopa inhibit the activity of *Chloracidobacterium thermophilum* PilB (*Ct*PilB). ATPase activity of *Ct*PilB was determined in triplicate in the presence of either (A) benserazide or (B) levodopa at the indicated concentrations. Activity without an inhibitor is normalized to 1.0. Each set of colored dots shows averages from one of three independent experiments (error bars omitted for clarity). The 50% inhibitory concentration ($IC_{50}$) determined by curve fitting is shown. Structures of benserazide and levodopa are shown in the insets of their respective panels.

in Fig. 3, the binding of PilB to MANT-ATP, and by extension to ATP, decreased with increasing concentrations of both inhibitors. Based on their effects on MANT-ATP binding, the $IC_{50}$s were calculated to be $33.6 \pm 0.5$ and $50.5 \pm 9.2$ $\mu$M for benserazide and levodopa, respectively. These results suggest that benserazide and levodopa are competitive inhibitors interfering with the binding of PilB to ATP.

*Ct*PilB, like its orthologue MshE (35), has a MshE$_N$-like cyclic di-GMP-binding domain at its N terminus and a well-conserved ATPase domain at its C terminus (28, 29). To determine which domain the inhibitors interact with, we tested the effects of benserazide and levodopa on the ATPase activity of a truncated *Ct*PilB variant (PilB$_{\Delta N}$) without the N-terminal MshE$_N$ domain (29). As shown in Fig. 4, both compounds inhibited the ATPase activity of PilB$_{\Delta N}$ in a concentration-dependent manner. The calculated $IC_{50}$ values against PilB$_{\Delta N}$ were $2.2 \pm 0.4$ and $76.2 \pm 9.2$ $\mu$M for benserazide and levodopa, respectively (Fig. 4A and B). These values are similar to those against the full-length PilB (Fig. 2). Combined with the observation that both inhibitors reduce the binding of an ATP analog (Fig. 3), these results suggest that benserazide and levodopa are competitive inhibitors, perhaps acting orthosterically. Assuming that they are competitive inhibitors, using their MANT-ATP $IC_{50}$ values, their inhibition constants ($K_i$) may be estimated by the Cheng-Prusoff equation (36, 37), resulting in a $K_i$ of 2.27 $\mu$M for benserazide and a $K_i$ of 3.40 $\mu$M for levodopa. While the mechanisms of inhibition have yet to be fully investigated, the results in Fig. 2 and 4 clearly demonstrate that benserazide and levodopa are effective inhibitors of the PilB ATPase *in vitro*.

**Benserazide and levodopa impede *M. xanthus* T4P-mediated motility.** Benserazide and levodopa were examined to determine whether they could impact the T4P assembly

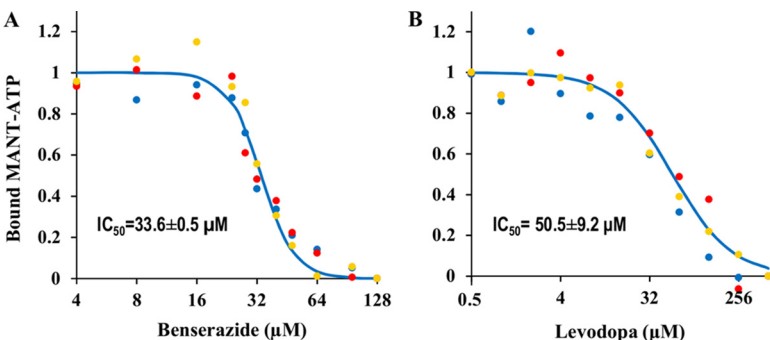

**FIG 3** Benserazide and levodopa reduce MANT-ATP binding to *Ct*PilB. Fluorescence was used to calculate the fraction of MANT-ATP bound to *Ct*PilB. Experiments were performed in triplicates with samples containing constant concentrations of *Ct*PilB and MANT-ATP with varying concentrations of (A) benserazide or (B) levodopa. The three different sets of colored dots represent averages from three independent experiments with error bars omitted for clarity. Best fits to isotherms with indicated $IC_{50}$s are shown.

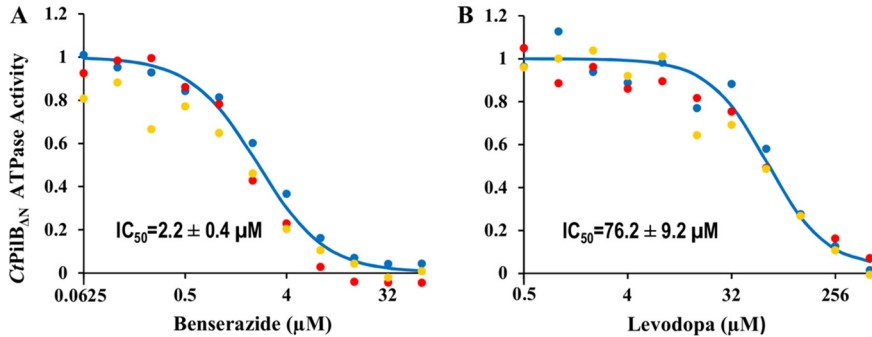

**FIG 4** Benserazide and levodopa inhibit the activity of $Ct$PilB$_{\Delta N}$. The ATPase activity of $Ct$PilB$_{\Delta N}$ was determined in the presence of benserazide (A) or levodopa (B) as described in Fig. 2.

function of PilB *in vivo* using *M. xanthus* as the model bacterium (28). This organism possesses T4P-powered social (S) motility, which can be analyzed on soft agar plates with low agar content (≤0.5%) (38, 39). Three *M. xanthus* strains were used for these experiments. Strain DK10416 lacks S motility because it is a *pilB* deletion (Δ*pilB*) mutant (40). The other two strains both assemble T4P and are proficient in S motility (30). Strain YZ1674 produces the wild-type (WT) *M. xanthus* PilB (*Mx*PilB). YZ2232 expresses *MC₃*PilB, a PilB chimera with the N terminus of *Mx*PilB fused to the C-terminal ATPase domain of *Ct*PilB (30). The PilB ATPase domains of *Mx*PilB and *MC₃*PilB are highly similar, with 66% shared identity and 77% similarity (Fig. S1).

Both benserazide and levodopa inhibited colony expansion of the two *M. xanthus* S-motile strains (YZ1674 and YZ2232) on soft agar plates (Fig. 5 and Fig. S2). Consistent with earlier observations *in vitro* (Fig. 2, 3, and 4), benserazide appeared to be a stronger inhibitor than levodopa. The former resulted in significant inhibition of S motility starting at 8 $\mu$M (Fig. 5A and C), the lowest concentration tested. In comparison, levodopa was less potent, as differences were observed only at ≥32 $\mu$M (Fig. 5B and D). Interestingly, despite the differences in the *pilB* genes of the two S-motile strains, their S motility showed similar responses to both inhibitors. It may be inferred that the two compounds inhibit *Mx*PilB and *Ct*PilB by interacting with features which are conserved among these two proteins (Fig. S1). The Δ*pilB* strain (DK10416), which was used as the negative control in this assay, showed no discernible expansion on the soft agar plate, as expected, because of its lack of S motility (Fig. S2, S3A and B).

Benserazide and levodopa could either inhibit S motility specifically or exhibit pleiotropic effects on *M. xanthus* growth or motility. *M. xanthus* possesses a T4P-independent motility known as adventurous (A) motility. We examined whether the expansion of the Δ*pilB* strain (DK10416), which possesses A motility only, was affected by the PilB inhibitors on hard agar plates which permit bacterial translocation by A and S motility (39). As shown in Fig. S3C and D, neither benserazide nor levodopa influenced the expansion of DK10416 on hard agar surfaces, indicating that they affected neither the A motility nor the growth of *M. xanthus* (see the Discussion section on growth in liquid culture). On the other hand, YZ1674 and YZ2232, which have both A and S motility, were affected in their expansion on hard agar by both compounds (Fig. S4 and S5) at concentrations similar to those on soft agar. Indeed, the results with hard agar plates mirror those with soft agar. That is, compared to the controls, statistically significant reductions were observed at ≥8 $\mu$M for benserazide and ≥32 $\mu$M for levodopa on both soft and hard agar surfaces. Because neither benserazide nor levodopa had any effect on A motility (Fig. S3C and D), the net differences on hard agar at different concentrations of the inhibitor (Fig. S4) can be viewed as confirmation of its effect on S motility (Fig. 5). Taken together, these results demonstrate that both benserazide and levodopa, which are PilB inhibitors *in vitro*, specifically inhibited *M. xanthus* S motility. It is plausible that this inhibition is attributable to their effects on the PilB ATPase *in vivo*.

**Benserazide and levodopa inhibit T4P assembly in *M. xanthus*.** These results on *M. xanthus* S motility (Fig. 5) indicate that benserazide and levodopa may inhibit PilB as the

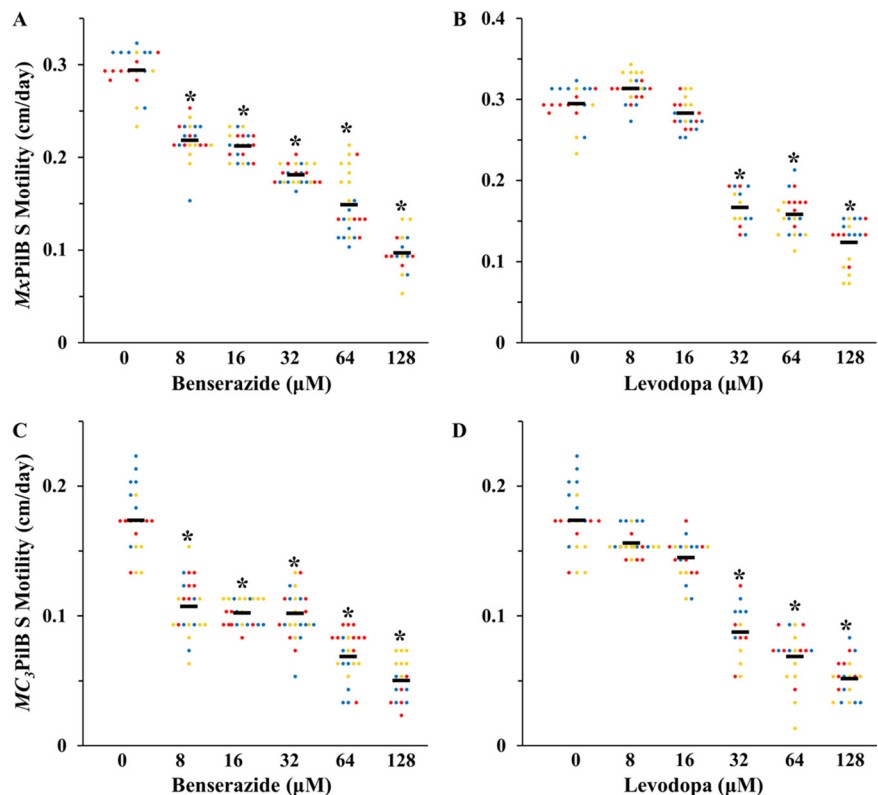

**FIG 5** Benserazide and levodopa inhibit T4P-mediated social (S) motility in *Myxococcus xanthus*. Expansion rates (cm/day) of the *Mx*PilB (YZ1674) and the *MC₃*PilB (YZ2232) strains on soft agar plates by S motility in the presence of (A and C) benserazide and (B and D) levodopa after 5 days of incubation. Results from three independent experiments are shown, each performed in quintuplicates with data points represented by dots of the same color. Horizontal black lines represent overall averages. Expansion rates were normalized by subtracting that of the Δ*pilB* strain, which has no S motility (also see Fig. S4). Asterisks (*) indicate that the values at the indicated concentrations are statistically different ($P < 0.05$) from those of the no-inhibitor control.

T4P assembly ATPase *in vivo*. Alternatively, they may inhibit other processes such as pilin production or T4P retraction. We next examined whether these two compounds reduced T4P assembly more specifically using a previously established protocol (28). Briefly, pre-existing pili were sheared off and separated from cells by vortexing and centrifugation. These shaved or unpiliated cells were allowed to assemble new T4P in the presence or absence of a PilB inhibitor. T4P fractions were prepared and analyzed by dot blotting with anti-pilin antibodies as a measure of T4P levels (28). These experiments were performed with the *M. xanthus* strain YZ603 (Δ*difE*), which assembles but does not retract its T4P (41–43). As such, the T4P fraction represents the level of T4P assembly without complication from T4P retraction (28).

The results shown in Fig. 6 demonstrate that both benserazide and levodopa reduce T4P assembly in *M. xanthus* in a concentration-dependent manner. Fig. 6A and B show a representative of the data from three biological experiments, each conducted with 4 technical replicates (R1 to R4). The complete data sets were analyzed and are shown in Fig. 6C and D. The results show that both benserazide and levodopa reduced T4P assembly in a dose-dependent manner. Consistent with earlier observations *in vitro* (Fig. 2, 3, and 4) and *in vivo* (Fig. 5 and Fig. S4), benserazide proved more potent than levodopa as an inhibitor of T4P assembly. For the former, statistically significant decreases in T4P assembly were observed at all concentrations, including the lowest, 16 μM. For the latter, significant reductions were observed at or above 32 μM. These results indicate that benserazide and levodopa both inhibit T4P assembly in *M. xanthus*, consistent with their inhibition of the PilB ATPase *in vivo*. Dot blotting of whole cells (WC) and other control fractions with anti-PilA antibodies showed that neither compound had any detectable effect on the pilin production level (Fig. S6B

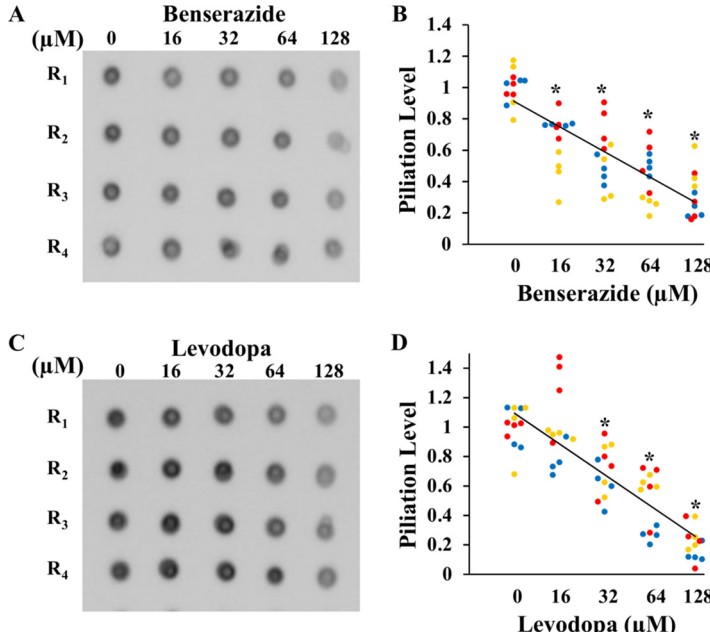

**FIG 6** *M. xanthus* T4P assembly is inhibited by benserazide and levodopa. Left side: representatives of dot blots of T4P fractions of cells treated with (A) benserazide or (C) levodopa at the indicated concentrations, with each sample analyzed in quadruplicates (R1 to R4), are shown. Right side: analysis of the complete data sets with benserazide (B) and levodopa (D) from three such experiments, with dots of the same color for data from the same experiment. Linear trendlines based on the averages are drawn for visualization purposes. Asterisks (*) indicate that piliation levels at the specified concentrations are statistically different ($P < 0.05$) from those of the untreated controls. Complete data sets for additional fractions and their analysis are shown in Fig. S6.

and D). These results indicate both benserazide and levodopa inhibit T4P assembly in the model bacterium *M. xanthus*, consistent with an effect on the PilB assembly ATPase *in vivo*.

**Benserazide and levodopa impede T4P-mediated motility and biofilm formation of *A. nosocomialis*.** These results suggest that benserazide and levodopa, which were identified as inhibitors of *Ct*PilB, are effective against *Mx*PilB, as well as against a PilB chimera with the ATPase core of *Ct*PilB *in vivo*. The inhibitors may thus interact with conserved features of the PilB ATPase family. *A. nosocomialis* PilB (19), whose N terminus is not similar to the MshE$_N$ domain (data not shown), shares ∼53% identity and ∼74% homology with *Mx*PilB and *MC₃*PilB in the ATPase domain (Fig. S1). We analyzed whether benserazide and levodopa could impact the T4P-powered twitching motility (19, 44) of *A. nosocomialis* (Fig. 7), an opportunistic pathogen in the *Acinetobacter calcoaceticus-Acinetobacter baumannii* (ACB) complex (45). As shown in Fig. 7A and B and Fig. S7, benserazide and levodopa impede the twitching motility of WT *A. nosocomialis* in a concentration-dependent manner. Both inhibitors significantly reduced *A. nosocomialis* twitching at ≥32 $\mu$M. At 128 $\mu$M, both compounds eliminated twitching of the WT strain such that it resembled the negative control (Δ*pilA*), which has no twitching (Fig. S7). Another T4P-related phenotype in *A. nosocomialis* is biofilm development, as the deletion of *pilA* was shown to abrogate *A. nosocomialis* biofilm formation (46). As shown in Fig. 7C and D, both compounds inhibited biofilm formation in a concentration-dependent manner. They both reduced biofilm formation significantly at concentrations of ≥32 $\mu$M, similar to their effective concentrations on *A. nosocomialis* twitching motility. These results provide additional evidence that both compounds disrupt T4P assembly in *A. nosocomialis*, suggesting that these inhibitors are effective against *A. nosocomialis* PilB *in vivo*.

## DISCUSSION

Here, we describe the implementation of an ATPase-based HTS for PilB inhibitors and the identification and validation of benserazide and levodopa as having inhibitory activities both *in vitro* and *in vivo*. We took advantage of the robust ATPase activity of *Ct*PilB *in vitro* (29, 30) to screen a library of 2,320 compounds for PilB inhibitors. This led us to benserazide and

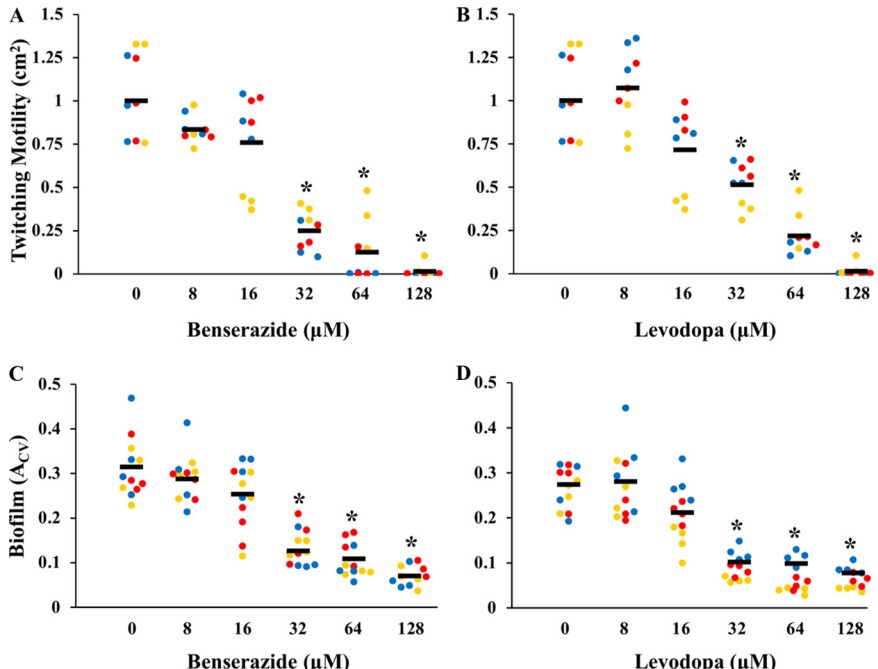

**FIG 7** Benserazide and levodopa inhibit *Acinetobacter nosocomialis* twitching motility and biofilm formation. (A and B) Twitching motility is correlated with the twitching area ($cm^2$) of the *A. nosocomialis* wild-type (WT) strain in the presence of (A) benserazide or (B) levodopa, with the control (no inhibitor) normalized to 1. Data are from 3 independent experiments, each performed in triplicates and represented by dots of the same color. (C and D) Biofilms formed by *A. nosocomialis* with (C) benserazide or (D) levodopa were analyzed by a microplate assay with crystal violet (CV) staining. Results from three independent experiments, each performed in quadruplicates and represented by dots of the same color, are shown. Horizontal lines indicate overall averages, and asterisks (*) signify that values at the indicated concentration are statistically different from those of the control ($P < 0.05$).

levodopa, two structurally related drugs that are approved for combination therapy against PD. We confirmed that their inhibitory activity is concentration-dependent, and our experimental evidence is consistent with their orthosteric inhibition of PilB. Results with the model organism *M. xanthus* indicated that these PilB inhibitors can inhibit T4P assembly and T4P-dependent motility. In addition, these compounds were shown to inhibit two T4P-dependent phenotypes in *A. nosocomialis*, namely, twitching motility and biofilm development. These results validate the use of *Ct*PilB as a model enzyme for the identification of PilB inhibitors against orthologous proteins in pathogenic and nonpathogenic bacteria alike. We note that the stepwise or multiphasic incline in the kinetic profile of *Ct*PilB described previously (30) could not be reproduced and it has been determined that it resulted from problems in data management and processing.

We had previously developed a screen for PilB inhibitors based on interference with the ability of *Ct*PilB to bind to the ATP analog MANT-ATP. Although this screen successfully identified quercetin from a small compound library, there were concerns that it heavily favored orthosteric inhibitors which directly compete with ATP binding. Considering the enzymatic actions of ATPases, it is expected that there are inhibitors which target other steps in the catalytic cycle, such as ATP hydrolysis and ADP release. These concerns motivated the development of this new HTS, which is based on ATPase activity. This screen would facilitate the identification of a fuller spectrum of PilB inhibitors, at least in theory. Surprisingly, both benserazide and levodopa were found to reduce the binding of PilB to MANT-ATP. They were also found to inhibit a *Ct*PilB variant containing only its C-terminal ATPase domains. These results are consistent with both benserazide and levodopa being orthosteric inhibitors. Other inhibitors from this screen (Fig. 1) have yet to be characterized and may have different modes of action, which would substantiate a preference for an ATPase activity-based HTS assay for the discovery of diverse PilB inhibitors over a screen based on displacement of MANT-ATP.

We argue that our results validate the approach of using *Ct*PilB as a model enzyme for the identification of PilB inhibitors for antivirulence purposes. There have been many attempts to biochemically characterize the PilB ATPase *in vitro*. The most success in this endeavor has been achieved with *Ct*PilB, which originates from the nonpathogenic, thermophilic bacterium *C. thermophilum* (29, 30). We used *Ct*PilB in a first HTS to identify quercetin as a PilB inhibitor (28). However, there have always been doubts as to whether *Ct*PilB could be reliably used as a model PilB for inhibitor identification for antivirulence in pathogenic bacteria. We were encouraged by the fact that many broad-spectrum antibiotics are effective against the same conserved cellular target in a wide range of bacteria. Penicillin, for example, is active against the penicillin-binding protein from diverse bacteria on the phylogenic tree. In this study, both benserazide and levodopa were initially identified and confirmed as *Ct*PilB inhibitors *in vitro*. Experiments with *M. xanthus* and the opportunistic pathogen *A. nosocomialis* indicate that they can both impact processes supported by PilB of *M. xanthus*, *C. thermophilum*, and *A. nosocomialis* origins. These observations, therefore, validated the approach of using the highly active *Ct*PilB as a model ATPase for the discovery of PilB inhibitors for exploration and studies of antivirulence and chemical biology.

The demonstration of benserazide and levodopa as being effective anti-T4P inhibitors *in vivo* lends hope to the development of antivirulence therapies against pathogens which use the T4P as a virulence factor. These include many bacteria with alarming prevalence for causing antibiotic-resistant infection (12). In this context, it is relevant that neither benserazide nor levodopa had any discernible effect on the growth of *M. xanthus*, *A. nosocomialis*, or *Escherichia coli* in liquid culture even at 512 $\mu$M, the highest concentration we tested (data not shown). This is important because, for the development of antivirulence chemotherapies, the small molecules should not have an inhibitory effect on bacterial growth. Otherwise, they would select for resistance, just like antibiotics. From a chemical structure perspective, levodopa is 3,4-dihydroxyphenylalanine (Fig. 2), or phenylalanine with two hydroxyl groups. We examined phenylalanine and tyrosine (4-monohydroxyphenylalanine) and found that they displayed no inhibitory effect on *Ct*PilB *in vitro* or the motility of *M. xanthus* and *A. nosocomialis in vivo* (data not shown). These preliminary studies of structure and activity relationships are consistent with benserazide and levodopa being specific inhibitors of PilB. As mentioned earlier, work with trifluoperazine (25) and P4MP4 (26, 27) has provided the proof of principle for antivirulence therapies with anti-T4P compounds. It is also encouraging that a compound targeting the adhesin function of the type 1 pilus of the uropathogenic *E. coli* (UPEC) (47) has moved to the clinical trial phase in the drug development pipeline (48).

A few of lines of evidence suggest that benserazide and levodopa act as competitive PilB inhibitors. First, both compounds reduced the binding of the ATP analogue MANT-ATP to PilB. Second, both inhibited the ATPase activity of a N-terminal truncated PilB with its conserved C-terminal ATPase core intact but without its MshE$_N$ domain. Finally, both inhibitors proved effective against T4P-mediated motility in two bacterial species with their PilB sharing homology only at the conserved C-terminal ATPase domains with variable N termini. These are all consistent with benserazide and levodopa acting as orthosteric PilB inhibitors. However, this simplistic interpretation of the mechanisms of inhibition must be taken with caution. There were two IC$_{50}$ values for each compound: one calculated from the effect on ATPase activity (Fig. 2), the other calculated from the effect on MANT-ATP binding (Fig. 3). The first values for the two compounds were 2.1 $\pm$ 0.4 $\mu$M and 58.8 $\pm$ 13.3 $\mu$M, respectively, which are significantly different. The second values, however, are much similar: 33.6 $\pm$ 0.5 $\mu$M and 50.5 $\pm$ 9.2 $\mu$M for benserazide and levodopa, respectively. In comparison, the two values for quercetin were 2.5 $\mu$M and 2.1 $\mu$M (28). These observations suggest that the modes of action for these inhibitors are likely more distinct than they are similar. It should be noted that PilB is a hexameric enzyme with six protomers in one functional unit. X-ray crystal structures indicated that these protomers are in at least 3 different conformations, which may correspond to different stages of the catalytic cycle in a ring-shaped enzyme with a rotary mechanism of catalysis (18). It remains to be determined whether or how these inhibitors interact with the different protomers in a hexamer to exert their inhibitory functions.

## MATERIALS AND METHODS

**HTS based on the malachite green ATPase assay and biochemical methods.** The full-length $Ct$PilB and the $Ct$PilB$_{\Delta N}$ variant (with the first 139 residues at the N terminus truncated) were purified as preciously described (29). The HTS was conducted using 384-well microplates (Greiner, cat no. 781101). Each microwell for the screen contained a 15-$\mu$L reaction mixture consisting of 75 nM $Ct$PilB, 0.5 mM ATP, 20 $\mu$M of a library compound, and 2% DMSO in activity buffer (50 mM $N$-tris(hydroxymethyl)methyl-3-aminopropanesulfonic acid [TAPS], 50 mM Tris, 75 mM KCl, 50 mM sodium acetate, 5 mM MgCl$_2$, 50 mM ZnCl$_2$ [pH 8.7]) (29). Reaction mixtures were incubated at 55°C for 10 min and transferred to ice for 5 min to stop the enzymatic reaction. MG reagent (50 $\mu$L) was added to each well for color development. After a 15-minute incubation, 5 $\mu$L of 34% (wt/vol) sodium-citrate (MP Chemicals) was added and the plate was mixed by shaking (28, 29). Absorbance at 620 nm ($A_{620}$) was measured using a SpectraMax M5 microplate reader at room temperature. The HTS was performed at the Virginia Tech Center for Drug Discovery (VTCDD) Screening Laboratory.

The effects of inhibitors on the ATPase activity of $Ct$PilB and $Ct$PilB$_{\Delta N}$ and the binding of MANT-ATP were determined as described previously (28, 29). Stock solutions were prepared by dissolving benserazide hydrochloride (Alfa Aesar) or levodopa (Spectrum Chemical Mfg. Corp.) in dimethyl sulfoxide (DMSO; Fisher Biotech). Next, 10× working stocks for given concentrations of an inhibitor were freshly prepared before their use in an experiment. GraphPad Prism v7.04 was used for curve fitting and Student's $t$ test was used for statistical analysis.

**Effects of benserazide and levodopa on $M.\ xanthus$ motility and T4P assembly.** $M.\ xanthus$ was grown and maintained at 32°C in Casitone-yeast extract (CYE) medium (49). Motility assays were conducted with strains DK10416 ($\Delta pilB$) (40), YZ1674 ($\Delta pilB\ att::MxpilB$) (50), and YZ2232 ($\Delta pilB\ att::MC_3pilB$) (30, 51) as previously described (28). One modification was that 10% India ink was included in the cell suspension to mark the initial spot on agar plates. All experiments were conducted in quintuplicates and the diameter of the initial spot, as marked by India ink, was subtracted for data analysis.

Piliation levels of the $M.\ xanthus$ strain YZ603 ($\Delta difE$) (52) were analyzed by dot-blotting with anti-PilA antibodies as previously described (28). This study included a fraction representing the "Basal" T4P levels for a given treatment. This is the T4P fraction from sheared cells mixed with an inhibitor and pre-equilibrated on ice for 10 min before the cells were incubated at 32°C for 20 min to allow T4P regrowth or assembly. These fractions were used as controls and subtracted as the baseline in data analysis.

**Examination of the effects of benserazide and levodopa on $A.\ nosocomialis$.** The $A.\ nosocomialis$ wild-type M2 strain and an isogenic $pilA$ deletion ($\Delta pilA$) mutant (19, 46) were used in this study. They were grown and maintained using MacConkey medium at 37°C. The twitching motility of the M2 strain of $A.\ nosocomialis$ was analyzed with 1% crystal violet (CV) for staining, and twitching areas were used as measures of motility as previously described (46). These motility assays were conducted in triplicates and the stained area of the $\Delta pilA$ mutant was subtracted from that of the WT for data analysis. $A.\ nosocomialis$ biofilm formation was analyzed using microplates and CV staining as previously described (53, 54). The biofilm assays were conducted in quadruplicate in MacConkey medium with tissue culture-treated microtiter plates (Falcon). CV absorbance ($A_{cv}$) was measured at 590 nm using Infinite M200 PRO in clear polystyrene 96-well microplates (ExtraGene). Absorbance for each well was read 16 times using Multiple Reads with the average as the final $A_{cv}$ value. Readings from control wells without bacteria were subtracted for normalization. Statistical differences were determined using a Student's $t$ test.

## SUPPLEMENTAL MATERIAL

Supplemental material is available online only.
**SUPPLEMENTAL FILE 1**, PDF file, 0.8 MB.

## ACKNOWLEDGMENTS

This work was partially supported by the National Science Foundation grants MCB-1417726 and MCB-1919455 and a Lay Nam Chang Dean's Discovery Fund to Z.Y. K.J.D. was the recipient of a GSDA and the Lewis Edward Goyette Fellowship, as well as the Liberati Scholarship from Virginia Tech.

We thank Kurt Piepenbrink for providing bacterial strains. We acknowledge Andreas Sukmana for his early contribution to the implementation of the HTS.

K.J.D., N.J.V., P.S., and Z.Y. designed research and analyzed data. K.J.D. and N.J.V. performed experiments. M.O. aided in figure preparations. W.S. and P.R.C. provided analysis of HTS results. K.J.D. and Z.Y. wrote the manuscript.

There are no conflicts of interest to declare.

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
