## [Reviewer comments · Microbiology Spectrum]

Microbiology Spectrum

Discovery of two inhibitors of the type IV pilus assembly ATPase PilB as potential antivirulence compounds

Keane Dye, Nancy Vogelaar, Megan O'Hara, Webster Santos, Pablo Sobrado, Paul Carlier, and Zhaomin Yang

Corresponding Author(s): Zhaomin Yang, Virginia Tech

Review Timeline:

Submission Date:	September 26, 2022
Editorial Decision:	October 14, 2022
Revision Received:	October 25, 2022
Editorial Decision:	October 26, 2022
Revision Received:	October 26, 2022
Accepted:	November 1, 2022

Editor: Paolo Visca

Reviewer(s): Disclosure of reviewer identity is with reference to reviewer comments included in decision letter(s). The following individuals involved in review of your submission have agreed to reveal their identity: Phoebe Lostroh (Reviewer #2)

Transaction Report:

DOI: <https://doi.org/10.1128/spectrum.03877-22>

October 14, 2022

Prof. Zhaomin Yang
Virginia Tech
Biological Sciences
117 Life Sciences I
970 Washington St. SW
Blacksburg, VA 24061-0910

Re: Spectrum03877-22 (Discovery of two inhibitors of the type IV pilus assembly ATPase as antivirulence compounds)

Dear Prof. Zhaomin Yang:

Thank you for submitting your manuscript to Microbiology Spectrum. The manuscript has been evaluated by two experts in the field. Both reviewers provided constructive criticism and recommended few useful revisions and some clarification. I agree with them, and I recommend you to revise the manuscript according to the suggestions of the two reviewers.

Link Not Available

Sincerely,

Paolo Visca

Journals Department
Reviewer comments:

Reviewer #1 (Comments for the Author):

In this manuscript the Authors report the identification of two PilB inhibitors, benserazide and levopoda, by a high throughput screen (HTS) in vitro, and their validation as effective inhibitors of type 4 pili (T4P) assembly in vivo. Notably, benserazide and levopoda inhibit T4P-dependent phenotypes, including different types of motility and biofilm formation, in *Myxococcus xanthus* and *Acinetobacter nosocomialis*.

In my opinion this is an interesting study, whose key findings are adequately supported by robust experimental data.

I have only few concerns that are detailed below:

- 1) Besides benserazide and levodopa, the identity of the additional 18 compounds identified as PilB inhibitors in the HTS campaign should be revealed.
- 2) The rationale behind the HTS system used in this study, that is based on the malachite green ATPase assay, should be clearly explained in the first chapter of the Results section.
- 3) The experiments performed to assess the specificity of benserazide and levodopa on S motility in *M. xanthus* are not clear to me. The rationale of these experiments is that in *M. xanthus* S motility is dependent on T4P and can be investigated on soft agar plates, while type A motility is T4P-independent and can be investigated on hard agar plates. Since the inhibitors did not affect type A motility in a *M. xanthus* mutant with pilB deletion (DK10416) on hard agar plates, the Authors conclude that benserazide and levodopa are specific inhibitors of T4P assembly and T4P-dependent motility (i.e. S motility). However, benserazide and levodopa also decreased motility of two *M. xanthus* pilB-proficient strains on hard agar plates, in a similar manner as previously observed on soft agar plates. Are T4P also required for motility on hard agar plates? Do hard agar plates allow investigating type A motility only, or both A and S type motility contribute to *M. xanthus* motility on hard agar plates? Please clarify.
- 4) In my opinion the title of the manuscript ("Discovery of two inhibitors of the type IV pilus assembly ATPase as antivirulence") does not conceive the main finding of this study, i.e. benserazide and levopoda inhibit the type IV pilus assembly ATPase PilB. Please consider that no direct evidences of the antivirulence activity of benserazide and levopoda have been produced in this study.

Minor comments

- 1) The acronym ABR is used only one time along the manuscript in place of antibiotic resistance (line 335). Please consider using antibiotic resistance also at line 335 and removing ABR.
- 2) Line 72: "pilT mutants in are hyperpilated", please correct.
- 3) Line 112: "(Greiner, 781101. Each", please add the parenthesis.
- 4) Line 114: Please define TAPS.
- 5) Line 115: "ZnCl₂", please correct.
- 6) Line 120: "performed by the at Virginia Tech", please correct.
- 7) Line 175: Please change Fig. 2A to Fig. 2.
- 8) Line 176: "withincreasing", please add space.

Reviewer #2 (Comments for the Author):

Title is appropriate

Rationale for the screen and execution of that screen are both sound. The use of *Chloro thermophilum* PilB is consistent with a previous publication in *mSphere* (2021)

Relevance of the results for curing infections in animals and ultimately humans would require much more work beyond the scope of this paper.

Figure 1: no comment

Figure 2: This comment applies to all the rest of the figures. Do not use red and green dots because those are not distinguishable for the most common forms of color blindness. Replace one of the dots with open (white) symbols and fill one of them with a pattern such as a stripe, or change the shape of one of the symbols to something else such as a triangle. It is good that the color scheme is conserved across figures.

Figure 2: X axis hard to read underneath the symbols after the 32 μ M point. Equation used to fit the line is appropriate.

Figure 3: No comment

Figure 4: no comment

Figure 5: I think it is odd to investigate the drugs' impact on Mx when Mx is not a pathogen, but their rationale for doing so is logical - I just would have preferred other assays using a pathogen instead. The subsequent use of Ano is more directly relevant to the ultimate goal of doing this study.

Figure 6: no comment

Figure 7: no comment

One minor concern: the drugs had no effect on the growth of Mx, Ano, or Eco in broth - but these drugs would presumably have stronger effects on biofilm formation. So, the measurement of growth in broth used to argue that this drug is less likely to lead to resolution is somewhat suspect.

The paper would be enhanced by using various concentrations of substrate in the presence of the drugs to establish the change in quantitative enzyme characteristics (needed to argue that the drugs are competitive inhibitors).

Staff Comments:

Preparing Revision Guidelines

Please return the manuscript within 60 days; if you cannot complete the modification within this time period, please contact me. If you do not wish to modify the manuscript and prefer to submit it to another journal, please notify me of your decision immediately so that the manuscript may be formally withdrawn from consideration by Microbiology Spectrum.

Response to Reviewers

We would like to thank both reviewers for your kind and constructive review of our manuscript. We greatly appreciate your expertise, time and efforts in providing timely and valuable comments and feedbacks on our manuscript. The peer review process would not work as intended otherwise.

Reviewer #1 (Comments for the Author):

In this manuscript the Authors report the identification of two PilB inhibitors, benserazide and levodopa, by a high throughput screen (HTS) in vitro, and their validation as effective inhibitors of type 4 pili (T4P) assembly in vivo. Notably, benserazide and levodopa inhibit T4P-dependent phenotypes, including different types of motility and biofilm formation, in *Myxococcus xanthus* and *Acinetobacter nosocomialis*.

In my opinion this is an interesting study, whose key findings are adequately supported by robust experimental data.

I have only few concerns that are detailed below:

1) Besides benserazide and levodopa, the identity of the additional 18 compounds identified as PilB inhibitors in the HTS campaign should be revealed.

Response: Judging by their chemical structures, some of 18 compounds are likely Pan-assay interference compounds known as PAINS in HTS-based drug discovery. A few others were found to be false positives which are also common in HTS. The remainders have either not been confirmed or not analyzed in any detail to be considered true PilB inhibitors with confidence. As a result, we feel it is better not to list the identities of these compounds to avoid disseminating misleading information.

2) The rationale behind the HTS system used in this study, that is based on the malachite green ATPase assay, should be clearly explained in the first chapter of the Results section.

Response: We added the following as the 2nd sentence in the first paragraph of the Results: “In this assay, phosphates from ATP hydrolysis and the MG reagents form green complexes that can be quantified calorimetrically with a plate reader”. We also provided additional information on the HTS in the following paragraph and provided the average Z' value for the screen.

3) The experiments performed to assess the specificity of benserazide and levodopa on S motility in *M. xanthus* are not clear to me. The rationale of these experiments is that in *M. xanthus* S motility is dependent on T4P and can be investigated on soft agar plates, while type A motility is T4P-independent and can be investigated on hard agar plates. Since the inhibitors did not affect type A motility in a *M. xanthus* mutant with pilB deletion (DK10416) on hard agar plates, the Authors conclude that benserazide and levodopa are specific inhibitors of T4P assembly and T4P-dependent motility (i.e. S motility). However, benserazide and levodopa also decreased motility of two *M. xanthus* pilB-proficient strains on hard agar plates, in a similar manner as previously observed on soft agar plates. Are T4P also required for motility on hard agar plates? Do hard agar plates allow investigating type A motility only, or both A and S type motility contribute to *M. xanthus* motility on hard agar plates? Please clarify.

Response: We have added information for clarification on line 231-234 to indicate that both A and S motilities are functional on hard agar plates. This is an important and thank you for pointing it out.

4) In my opinion the title of the manuscript ("Discovery of two inhibitors of the type IV pilus assembly ATPase as antivirulence") does not conceive the main finding of this study, i.e. benserazide and levopoda inhibit the type IV pilus assembly ATPase PilB. Please consider that no direct evidences of the antivirulence activity of benserazide and levopoda have been produced in this study.

Response: We have inserted PilB in the title as we agree that it is helpful. We have left the remainder of the title as is considering the comment by the 2nd reviewers that the "Title is appropriate".

Minor comments

1) The acronym ABR is used only one time along the manuscript in place of antibiotic resistance (line 335). Please consider using antibiotic resistance also at line 335 and removing ABR.

2) Line 72: "pilT mutants in are hyperpiliated", please correct.

3) Line 112: "(Greiner, 781101. Each", please add the parenthesis.

4) Line 114: Please define TAPS.

5) Line 115: "ZnCl2", please correct.

6) Line 120: "performed by the at Virginia Tech", please correct.

7) Line 175: Please change Fig. 2A to Fig. 2.

8) Line 176: "withincreasing", please add space.

Response: All the above typo/grammatical errors have been corrected.

Reviewer #2 (Comments for the Author):

Title is appropriate

Rationale for the screen and execution of that screen are both sound. The use of Chloro thermophilum PilB is consistent with a previous publication in mSphere (2021)

Relevance of the results for curing infections in animals and ultimately humans would require much more work beyond the scope of this paper.

Figure 1: no comment

Figure 2: This comment applies to all the rest of the figures. Do not use red and green dots because those are not distinguishable for the most common forms of color blindness. Replace one of the dots with open (white) symbols and fill one of them with a pattern such as a stripe, or change the shape of one of the symbols to something else such as a triangle . It is good that the color scheme is conserved across figures.

Response: We replaced the green color so that a colorblind person could tell the different of the three color in all the figures.

Figure 2: X axis hard to read underneath the symbols after the 32 μM point. Equation used to fit the line is appropriate.

Response: We lowered the X intercept with Y so that the labeling is no longer obstructed by the graph.

Figure 3: No comment

Figure 4: no comment

Figure 5: I think it is odd to investigate the drugs' impact on Mx when Mx is not a pathogen, but their rationale for doing so is logical - I just would have preferred other assays using a pathogen instead. The subsequent use of Ano is more directly relevant to the ultimate goal of doing this study.

Response: We appreciate the recognition of the logic for using Mx as a model organism. As noted by the reviewer, we used the pathogen *A. nosocomialis* in experiments described later in the manuscript as the work progressed along.

Figure 6: no comment

Figure 7: no comment

One minor concern: the drugs had no effect on the growth of Mx, Ano, or Eco in broth - but these drugs would presumably have stronger effects on biofilm formation. So, the measurement of growth in broth used to argue that this drug is less likely to lead to resolution is somewhat suspect.

Response: The experiments were to examine if these drugs have any antimicrobial activity. Broth cultures are typically used to determine the minimum inhibitory concentration (MIC) of antimicrobials.

The paper would be enhanced by using various concentrations of substrate in the presence of the drugs to establish the change in quantitative enzyme characteristics (needed to argue that the drugs are competitive inhibitors).

Response: We agree that understanding the mechanisms of inhibition by these drugs are important. We consider that the determination of the precise mechanisms is beyond the scope of this manuscript.

October 26, 2022

Prof. Zhaomin Yang
Virginia Tech
Biological Sciences
117 Life Sciences I
970 Washington St. SW
Blacksburg, VA 24061-0910

Re: Spectrum03877-22R1 (Discovery of two inhibitors of the type IV pilus assembly ATPase PilB as antivirulence compounds)

Dear Prof. Zhaomin Yang:

Thank you for submitting your manuscript to Microbiology Spectrum. As you will see your paper is very close to acceptance. Please modify the manuscript along the lines I have recommended. As these revisions are quite minor, I expect that you should be able to turn in the revised paper soon. If your manuscript was reviewed, you will find the reviewers' comments below.

My Editor's comment to your revised manuscript is:

I agree with comment 4 of Reviewer 1, as no experimental data are provided to confirm virulence inhibition. This issue should be addressed by adequate cellular and animal models, that exceed the scope of this manuscript. Therefore, I recommend more caution in the title, that should be slightly modified as follows:

"Discovery of two inhibitors of the type IV pilus assembly ATPase PilB as potential antivirulence compounds".

Please, notice that the acceptance of the title by Reviewer 2 does not imply that the comment of Reviewer 1 is inconsistent.

When submitting the revised version of your paper, please provide (1) point-by-point responses to the issues raised by the reviewers as file type "Response to Reviewers," not in your cover letter, and (2) a PDF file that indicates the changes from the original submission (by highlighting or underlining the changes) as file type "Marked Up Manuscript - For Review Only". Please use this link to submit your revised manuscript. Detailed instructions on submitting your revised paper are below.

Link Not Available

Sincerely,

Paolo Visca

Reviewer comments:

None

Preparing Revision Guidelines

- Point-by-point responses to the issues raised by the reviewers in a file named "Response to Reviewers," NOT IN YOUR COVER LETTER.
- Upload a compare copy of the manuscript (without figures) as a "Marked-Up Manuscript" file.
- Each figure must be uploaded as a separate file, and any multipanel figures must be assembled into one file.
- Manuscript: A .DOC version of the revised manuscript

- Figures: Editable, high-resolution, individual figure files are required at revision, TIFF or EPS files are preferred

Please return the manuscript within 60 days; if you cannot complete the modification within this time period, please contact me. If you do not wish to modify the manuscript and prefer to submit it to another journal, please notify me of your decision immediately so that the manuscript may be formally withdrawn from consideration by Microbiology Spectrum.

Response to Reviewers

We would like to thank both reviewers for your kind and constructive review of our manuscript. We greatly appreciate your expertise, time and efforts in providing timely and valuable comments and feedbacks on our manuscript. The peer review process would not work as intended otherwise.

Reviewer #1 (Comments for the Author):

In this manuscript the Authors report the identification of two PilB inhibitors, benserazide and levodopa, by a high throughput screen (HTS) in vitro, and their validation as effective inhibitors of type 4 pili (T4P) assembly in vivo. Notably, benserazide and levodopa inhibit T4P-dependent phenotypes, including different types of motility and biofilm formation, in *Myxococcus xanthus* and *Acinetobacter nosocomialis*.

In my opinion this is an interesting study, whose key findings are adequately supported by robust experimental data.

I have only few concerns that are detailed below:

1) Besides benserazide and levodopa, the identity of the additional 18 compounds identified as PilB inhibitors in the HTS campaign should be revealed.

Response: Judging by their chemical structures, some of 18 compounds are likely Pan-assay interference compounds known as PAINS in HTS-based drug discovery. A few others were found to be false positives which are also common in HTS. The remainders have either not been confirmed or not analyzed in any detail to be considered true PilB inhibitors with confidence. As a result, we feel it is better not to list the identities of these compounds to avoid disseminating misleading information.

2) The rationale behind the HTS system used in this study, that is based on the malachite green ATPase assay, should be clearly explained in the first chapter of the Results section.

Response: We added the following as the 2nd sentence in the first paragraph of the Results: “In this assay, phosphates from ATP hydrolysis and the MG reagents form green complexes that can be quantified calorimetrically with a plate reader”. We also provided additional information on the HTS in the following paragraph and provided the average Z' value for the screen.

3) The experiments performed to assess the specificity of benserazide and levodopa on S motility in *M. xanthus* are not clear to me. The rationale of these experiments is that in *M. xanthus* S motility is dependent on T4P and can be investigated on soft agar plates, while type A motility is T4P-independent and can be investigated on hard agar plates. Since the inhibitors did not affect type A motility in a *M. xanthus* mutant with pilB deletion (DK10416) on hard agar plates, the Authors conclude that benserazide and levodopa are specific inhibitors of T4P assembly and T4P-dependent motility (i.e. S motility). However, benserazide and levodopa also decreased motility of two *M. xanthus* pilB-proficient strains on hard agar plates, in a similar manner as previously observed on soft agar plates. Are T4P also required for motility on hard agar plates? Do hard agar plates allow investigating type A motility only, or both A and S type motility contribute to *M. xanthus* motility on hard agar plates? Please clarify.

Response: We have added information for clarification on line 231-234 to indicate that both A and S motilities are functional on hard agar plates. This is an important and thank you for pointing it out.

4) In my opinion the title of the manuscript ("Discovery of two inhibitors of the type IV pilus assembly ATPase as antivirulence") does not conceive the main finding of this study, i.e. benserazide and levopoda inhibit the type IV pilus assembly ATPase PilB. Please consider that no direct evidences of the antivirulence activity of benserazide and levopoda have been produced in this study.

Response: We have inserted PilB in the title as we agree that it is helpful. We have left the remainder of the title as is considering the comment by the 2nd reviewers that the "Title is appropriate".

Minor comments

1) The acronym ABR is used only one time along the manuscript in place of antibiotic resistance (line 335). Please consider using antibiotic resistance also at line 335 and removing ABR.

2) Line 72: "pilT mutants in are hyperpiliated", please correct.

3) Line 112: "(Greiner, 781101. Each", please add the parenthesis.

4) Line 114: Please define TAPS.

5) Line 115: "ZnCl2", please correct.

6) Line 120: "performed by the at Virginia Tech", please correct.

7) Line 175: Please change Fig. 2A to Fig. 2.

8) Line 176: "withincreasing", please add space.

Response: All the above typo/grammatical errors have been corrected.

Reviewer #2 (Comments for the Author):

Title is appropriate

Rationale for the screen and execution of that screen are both sound. The use of Chloro thermophilum PilB is consistent with a previous publication in mSphere (2021)

Relevance of the results for curing infections in animals and ultimately humans would require much more work beyond the scope of this paper.

Figure 1: no comment

Figure 2: This comment applies to all the rest of the figures. Do not use red and green dots because those are not distinguishable for the most common forms of color blindness. Replace one of the dots with open (white) symbols and fill one of them with a pattern such as a stripe, or change the shape of one of the symbols to something else such as a triangle . It is good that the color scheme is conserved across figures.

Response: We replaced the green color so that a colorblind person could tell the different of the three color in all the figures.

Figure 2: X axis hard to read underneath the symbols after the 32 μM point. Equation used to fit the line is appropriate.

Response: We lowered the X intercept with Y so that the labeling is no longer obstructed by the graph.

Figure 3: No comment

Figure 4: no comment

Figure 5: I think it is odd to investigate the drugs' impact on Mx when Mx is not a pathogen, but their rationale for doing so is logical - I just would have preferred other assays using a pathogen instead. The subsequent use of Ano is more directly relevant to the ultimate goal of doing this study.

Response: We appreciate the recognition of the logic for using Mx as a model organism. As noted by the reviewer, we used the pathogen *A. nosocomialis* in experiments described later in the manuscript as the work progressed along.

Figure 6: no comment

Figure 7: no comment

One minor concern: the drugs had no effect on the growth of Mx, Ano, or Eco in broth - but these drugs would presumably have stronger effects on biofilm formation. So, the measurement of growth in broth used to argue that this drug is less likely to lead to resolution is somewhat suspect.

Response: The experiments were to examine if these drugs have any antimicrobial activity. Broth cultures are typically used to determine the minimum inhibitory concentration (MIC) of antimicrobials.

The paper would be enhanced by using various concentrations of substrate in the presence of the drugs to establish the change in quantitative enzyme characteristics (needed to argue that the drugs are competitive inhibitors).

Response: We agree that understanding the mechanisms of inhibition by these drugs are important. We consider that the determination of the precise mechanisms is beyond the scope of this manuscript.

November 1, 2022

Prof. Zhaomin Yang
Virginia Tech
Biological Sciences
117 Life Sciences I
970 Washington St. SW
Blacksburg, VA 24061-0910

Re: Spectrum03877-22R2 (Discovery of two inhibitors of the type IV pilus assembly ATPase PilB as potential antivirulence compounds)

Dear Prof. Zhaomin Yang:

Your manuscript has been accepted, and I am forwarding it to the ASM Journals Department for publication. You will be notified when your proofs are ready to be viewed.

Sincerely,

Paolo Visca
Editor, Microbiology Spectrum
